# Influence of a Non-Resonant Intense Laser and Structural Defect on the Electronic and Optical Properties of a GaAs Quantum Ring under Inversely Quadratic Potential

José C. León-González [1,2,*], Rafael G. Toscano-Negrette [1,2], Juan A. Vinasco [1], Alvaro L. Morales [1], Miguel E. Mora-Ramos [3] and Carlos A. Duque [1]

[1] Grupo de Materia Condensada-UdeA, Instituto de Física, Facultad de Ciencias Exactas y Naturales, Universidad de Antioquia UdeA, Calle 70 No. 52-21, Medellín 050010, Colombia; rafael.toscano@udea.edu.co (R.G.T.-N.); juan.vinascos@udea.edu.co (J.A.V.); alvaro.morales@udea.edu.co (A.L.M.); carlos.duque1@udea.edu.co (C.A.D.)

[2] Departamento de Física y Electrónica, Universidad de Córdoba, Carrera 6 No. 77-305, Montería 23002, Colombia

[3] Centro de Investigación en Ciencias, Instituto de Investigación en Ciencias Básicas y Aplicadas, Universidad Autónoma del Estado de Morelos, Av. Universidad 1001, Cuernavaca 62209, Mexico; memora@uaem.mx

* Correspondence: jose.leong@udea.edu.co

**Abstract:** We investigated the impact of a non-resonant intense laser, structural defects, and magnetic fields on the electronic and optical properties of a simple GaAs quantum ring under the inverse quadratic Hellmann potential, using the effective mass and parabolic band approximations. We obtained the energies and wavefunctions by solving the 2D Schrodinger's equation using the finite-element numerical technique to analyze this. We considered circular polarization to calculate the dipole matrix elements, which were influenced by the laser field and structural defects in the system. This enabled us to study the linear absorption coefficients. Our results demonstrated that the presence of a laser field and a structural defect disrupt the axial symmetry of the problem. When only the non-resonant laser was present, a pattern of excited states appeared in pairs, which oscillated with the magnetic field. However, the amplitude of the oscillation decreased as the magnetic field strength increased, and these oscillations disappeared when the structural defect was introduced. It was also noted that the intensity and position of the linear optical absorption peaks exhibited a non-monotonic behavior with the magnetic field in the absence of a structural defect. However, this behavior changed when the structural defect was present, depending on the type of polarization (right or left circular). Finally, a clear improvement in the absorption peaks with an increase in the laser parameter is reported.

**Keywords:** non-resonant intense laser; structural defect; circular polarization; linear absorption coefficients

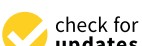



## 1. Introduction

The study of low-dimensional semiconductor structures has gained significant interest in recent decades due to their unique and distinct properties compared to conventional structures. These properties offer various potential applications, including electronic devices, solar cells, sensors, etc. One such structure is the quantum ring (QR). Similar to the so-called quantum dots, the QRs are quasi-zero-dimensional structures that completely confine the motion of charge carriers, with the difference of being doubly connected from the topological point of view. These structures can be designed in both two (with a disc-like shape) and three dimensions. Given the specific features of semiconductor QRs, it is possible to foresee their use in fabricating optoelectronic devices with targeted functions.

The electronic and optical properties of charge carriers in QRs can be adjusted by various external factors such as size, geometry, hydrostatic pressure, temperature, impurities, electric and magnetic fields, structural defects, and intense resonant and non-resonant laser fields, among others [1–29]. All these factors would allow optimizing the devices manufactured with these semiconductor nanostructures. Overall, the study of low-dimensional semiconductor structures, in particular QRs, has tremendous potential for the development of new and innovative applications in numerous fields [30].

The present work was devoted, in the first place, to investigating the spectrum of energy levels in a semiconductor two-dimensional QR whose design included a particular description of the confining potential function and the possibility of having a structural defect. In addition, the study included an analysis of the influence of external electromagnetic probes, such as a static magnetic field and a non-resonant intense laser field, on electron states. This implies determining the solution of the conduction-band effective mass equation under the stated conditions. Secondly, the information from the allowed quantum states will serve as the basis for the calculation of one of the possible optical responses associated with transitions between them: the coefficient of intersubband light absorption (here, as it is customary, we shall keep the term "intersubband", although no band is actually present in the electronic structure of the system).

Among the vast number of reports regarding the influence of applied magnetic fields on the properties of QRs, we shall refer only to a few for illustration. For instance, in one of the earlier works, Barticevic et al. analyzed these properties by including different potential models of ring confinement and distinct geometric configurations. They found that a magnetic field causes a notable enhancement of the resonance intensities in the optical response [31]. Li and Peeters investigated the tunable optical Aharonov–Bohm effect in a semiconductor QR [32]. On the other hand, a model for a two-dimensional QR with an inverse square potential and the presence of a magnetic field was presented in [33]. There, the nonlinear optical properties associated with inter-level transitions were investigated.

Non-resonant intense laser fields offer another way to control the properties of QRs. Such fields excite the charge carrier in the ring non-resonantly, meaning that the laser field's energy does not match any of the possible energy transitions of the confined electron in the ring. The average temporal effect of the non-resonant laser field is to produce changes in the confinement potential, which confines the electron into the heterostructure region. This interaction can result in interesting effects, such as the generation of harmonics [20–24] and the shift in the position of the optical absorption peaks at higher energies as the intensity of the laser field increases [25,26].

The presence of a structural defect in the system can also lead to significant changes in nanosystems' electronic and optical structure. For example, Castaño-Yepes et al. [27] studied the impact of this kind of geometric feature on the thermomagnetic and optical properties of a two-dimensional quantum dot. They found that introducing a conical declination modifies the selection rules for studying optical transitions, creating a new set of allowed transitions. Furthermore, they discovered that the parameter controlling the degree of this structural defect affects the thermodynamic quantities, playing a crucial role in determining the values of the thermal response functions [28]. It should also be noted that the presence of a conical declination leads to symmetry breaking of the problem [29].

Taking into account all the above, the goal of this work was to theoretically investigate the impact of a non-resonant intense laser and a structural defect in the form of a circular sector cut on the electronic and optical characteristics of a two-dimensional GaAs QR containing a confined electron. The carrier was assumed to be under the influence of external magnetic fields and an inversely quadratic Hellmann-type confinement potential. From the electronic point of view, the task consisted of obtaining the energies and eigenfunctions of the system through the solution of the effective mass Schrödinger equation using the finite-element method (FEM) implemented in the Comsol-Multiphysics software. For that purpose, the parabolic band approximation was used. Quantities such as magnetic field intensity and the intense laser field parameter were varied accordingly. Transition energies

between the ground and first two excited states were evaluated and, together with the associated electric dipole moment matrix elements, were employed in calculating the linear optical absorption coefficient for electrons in the system. This process was carried out for both right and left circular polarizations of incident light. The paper is structured as follows: Section 2 describes the theoretical model; Section 3 presents the results and discussion; Section 4 summarizes the essential findings and conclusions.

## 2. Theoretical Model

This study investigated the influence of magnetic fields, non-resonant intense laser fields, and structural defects on an electron's electronic and optical properties in a QR subjected to the inverse quadratic Hellmann potential. Using the effective mass approximation, we can express the Schrödinger equation as follows:

$$\left[ \frac{1}{2m^*} \left( \vec{p} + e\,\vec{A} \right)^2 + V(x,y) \right] \psi_n(x,y) = E_n\,\psi_n(x,y), \tag{1}$$

where $m^*$ denotes the effective mass of the electron in GaAs, $e$ represents the absolute value of the electron charge, $\vec{A} = \frac{B}{2}(-y, x, 0)$ is the vector potential defined using the symmetric gauge, $B$ is the magnitude of the magnetic field (which is directed in the $+z$-direction in this study), and $V(x,y)$ is the 2D confinement potential. As mentioned, the confinement potential was modeled using the inversely quadratic Hellmann potential, which is given by [34]

$$V(x,y) = V_0 \left[ \frac{1}{2} - \frac{2\,R_0}{\sqrt{x^2 + y^2}} + \left( \frac{R_0}{\sqrt{x^2 + y^2}} \right)^2 \right], \tag{2}$$

where $V_0$ is the potential barrier height and $R_0$ is the QR radius. The minimum value of this potential is found at $R_0$, with a minimum energy of $-V_0/2$. If observed from the radial point of view, the potential tends to $+V_0/2$ when $r = \sqrt{x^2 + y^2}$ tends to infinity. Therefore, the active potential region is between $-V_0/2$ and $+V_0/2$. From a practical point of view, this type of potential is achieved by an intentional variation of the aluminum concentration along the radial direction of the heterostructure. This is a challenging issue and quite complex to achieve. However, it is important to highlight that these non-abrupt variations of the confinement potentials occur in the interdiffusion process between regions of wells and barriers when these are intended to be built abruptly. At this point, the model that we present here takes on its greatest relevance.

Including a non-resonant laser field in the system affects the shape and intensity of confinement potential energy. Consequently, it is referred to as a dressed potential, and its new mathematical form can be written as [35,36]

$$V_d(x,y) = \frac{1}{2\pi} \int_0^{2\pi} V(X(t), Y(t))\,dt, \tag{3}$$

where $X(t) = x + \alpha_0 \cos(\beta) \cos(t)$, $Y(t) = y + \alpha_0 \sin(\beta) \cos(t)$, and $\alpha_0$ are the parameters that control the laser intensity, and $\beta$ is the laser polarization. In this work, we shall consider linear polarization along the +x axis; therefore, $\beta = 0$.

Figure 1 shows the plots of the original potential (black line) and the one dressed by the laser with different values of $\alpha_0$. It is possible to notice the important effect that the laser generates on the confinement, shifting upwards the well bottom and reducing the height of the potential barrier. Consequently, the states are less confined to what is expected. Furthermore, a change occurs in the position of the minimum where it is now greater than $R_0$.

For this work, the effects of including a structural defect in the system's geometry were also considered. This consisted of extracting a circular disk sector that represents the geometry. Then, Figure 2 shows the geometry of the structure, where an inner radius $R_0$ is defined, which is the radius of the ring, and an outer radius $R_1$ to establish the boundary

conditions; Figure 2a,b show the cases without and with structural defects, respectively. The corresponding mesh used in FEM calculations is depicted as well. Note that the angle $\theta_0$ represents the remaining angular section, which was 350° for this work. Thus, the angular amplitude of the structural defect introduced in the problem was 10°.

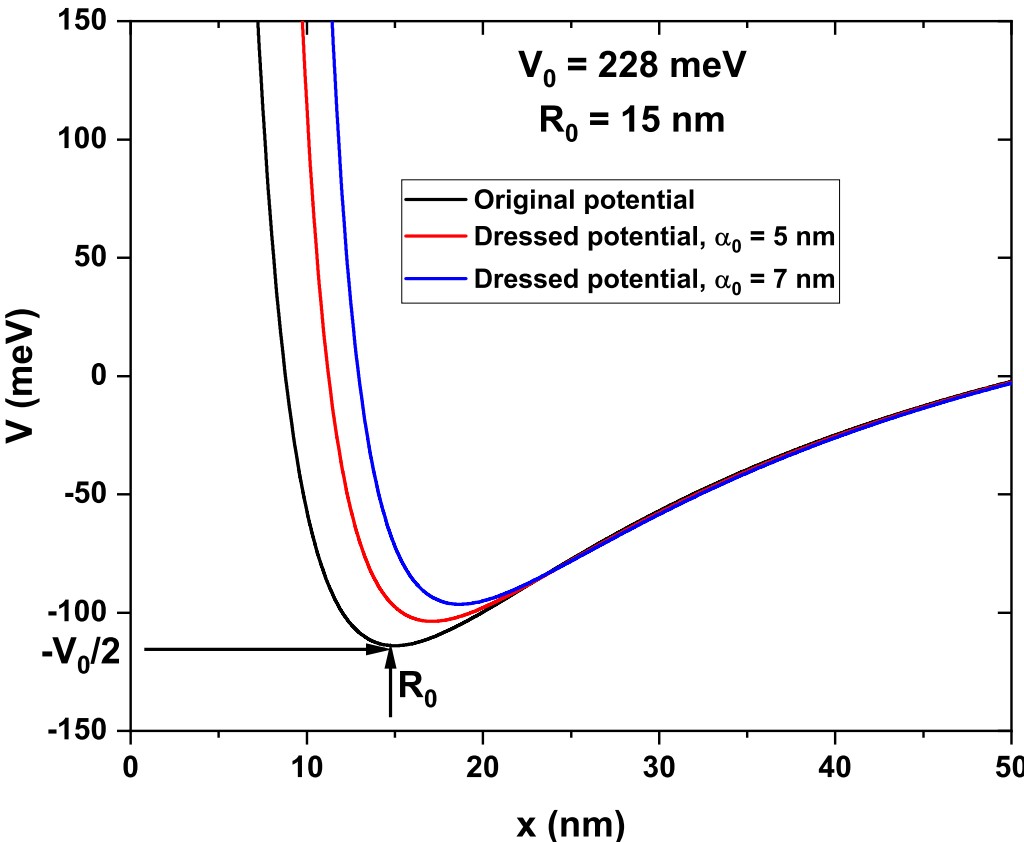

**Figure 1.** Inversely quadratic potential (Hellmann potential) as a function of the spatial coordinate, *x*, undressed (black line) and dressed by the laser (red and blue lines) with two values of the laser parameter.

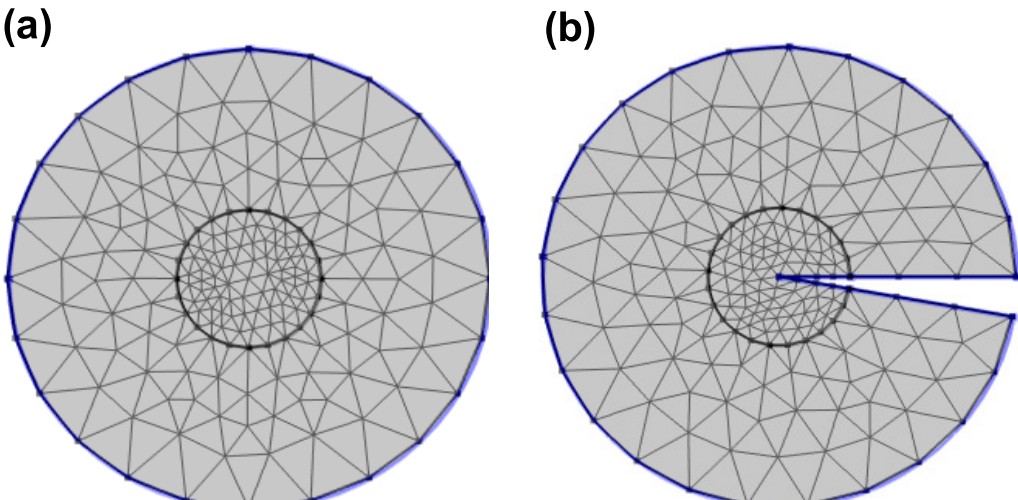

**Figure 2.** Structure geometry without structural defect (**a**) and with structural defect (**b**). The black lines are the rectangular mesh used for this problem, where double refinement was used in the inner circular region of radius $R_0$, and the blue line corresponds to the boundary conditions established for this problem.

When the system energies and wavefunctions have been obtained by solving Equation (1), the features of intersubband transitions are reflected in the linear optical absorption coefficient, which is evaluated using the following expression:

$$\alpha^{(1)}(\omega) = \sqrt{\frac{\mu_0}{\epsilon_r \epsilon_0}} \sum_{j=2}^{j=3} \frac{\omega \, e^2 \, \sigma \, \hbar \, \Gamma |M_{1j}|^2}{(E_{1j} - \hbar \, \omega)^2 + (\hbar \, \Gamma)^2},$$

(4)

where $\omega$ is the frequency of the incident photon, $\sigma$ the carrier density, $\Gamma$ the carrier damping rate in the transition, $|M_{1j}|^2$ the square of the dipole moment, and $E_{1j}$ the transition energy. We studied the optical properties considering the transitions between the ground state and the first and second excited states, denoted as $E_{12} = E_2 - E_1$ and $E_{13} = E_3 - E_1$, respectively. Therefore, in each case, the resonant peaks were found at photon energies close to $E_{1j}$. The square of the dipole moment $|M_{1j}|^2$ can be written as

$$|M_{1j}| = \int \int \psi_j^* \, \zeta \, \psi_1 \, dx \, dy,$$

(5)

where $\zeta$ is the incident light polarization. For this work, the incoming electromagnetic signal was considered circularly polarized light, defined as right circular polarization $\zeta = \frac{x+iy}{\sqrt{2}}$ and left circular polarization $\zeta = \frac{x-iy}{\sqrt{2}}$.

As mentioned above, the Schrödinger equation, defined in Equation (1), was solved by the FEM through the COMSOL-Multiphysics [37] software. For both cases (without and with a structural defect), a rectangular mesh with double refinement was used in the region $0 < r < R_0$ due to the divergence of the potential when $r \to 0$. In the absence of a structural defect, the mesh details were 1554 mesh vertices, 3046 triangles, 156 edge elements, 8 vertex elements, 3046 elements, a 0.3202 quality element minimum, a 0.8152 quality element medium, a 0.01373 element area ratio, and a 7840 nm$^2$ mesh area. When considering the structural defect, the parameters were 1322 mesh vertices, 2544 triangles, 178 edge elements, 11 vertex elements, 2544, a 0.3259 quality element minimum, a 0.8158 quality element medium, a 0.01969 element area ratio, and a 7623 nm$^2$ mesh area.

## 3. Results and Discussion

With the purpose of determining the electronic and optical properties of the electron confined in the case of a GaAs QR under the effects of the intense non-resonant laser field and structural defect, the following parameters were used: $R_0 = 15$ nm, $R_1 = 50$ nm, $m^* = 0.067 \, m_0$, $V_0 = 228$ meV, $\sigma = 3 \times 10^{22}$ m$^{-3}$, $\hbar \, \Gamma = 0.5$ meV, and $\epsilon_r = 12.58$ [38,39].

The aim was to examine the behavior of energy states in the conduction band for an electron confined in the QR, when a magnetic field was applied to the system. Figure 3 illustrates the corresponding variations of the lowest seven energy levels for different values of the laser parameter $\alpha_0$, as functions of the magnetic field applied perpendicular to the plane. When $\alpha_0 = 0$ (Figure 3a), the system exhibited axial symmetry, resulting in doubly degenerate excited states for $B = 0$. However, this degeneracy was broken when a magnetic field was applied, and multiple crossings between energy levels occurred for particular magnetic field values, leading to accidental degeneracies. This resulted in an oscillating pattern of energies as a function of $B$, and at the crossing points, the corresponding wave function changed symmetry. When the laser parameter was activated (see Figure 3b,c), the energy levels behaved differently, and symmetry breaking was noted in the structure. The lowest excited states lost their degeneracy even when $B = 0$, and the crossing points between the levels occurred for smaller fields than in Figure 3a. It is worth mentioning that the oscillating pattern between the energy levels was still observed, but it was now organized in pairs with smaller amplitudes for the first two levels and larger amplitudes for the upper ones. These amplitudes decreased as the laser parameter increased, as shown in Figure 3a. The boxes in Figure 3a–c depict the ground state wavefunction for each case. It was evident that, with the application of the laser field, the wave function was composed

of two well-defined lobes when the laser parameter was $\alpha_0 = 7\,\text{nm}$. Thus, the electron can occupy these two isolated—and well-defined—regions of space.

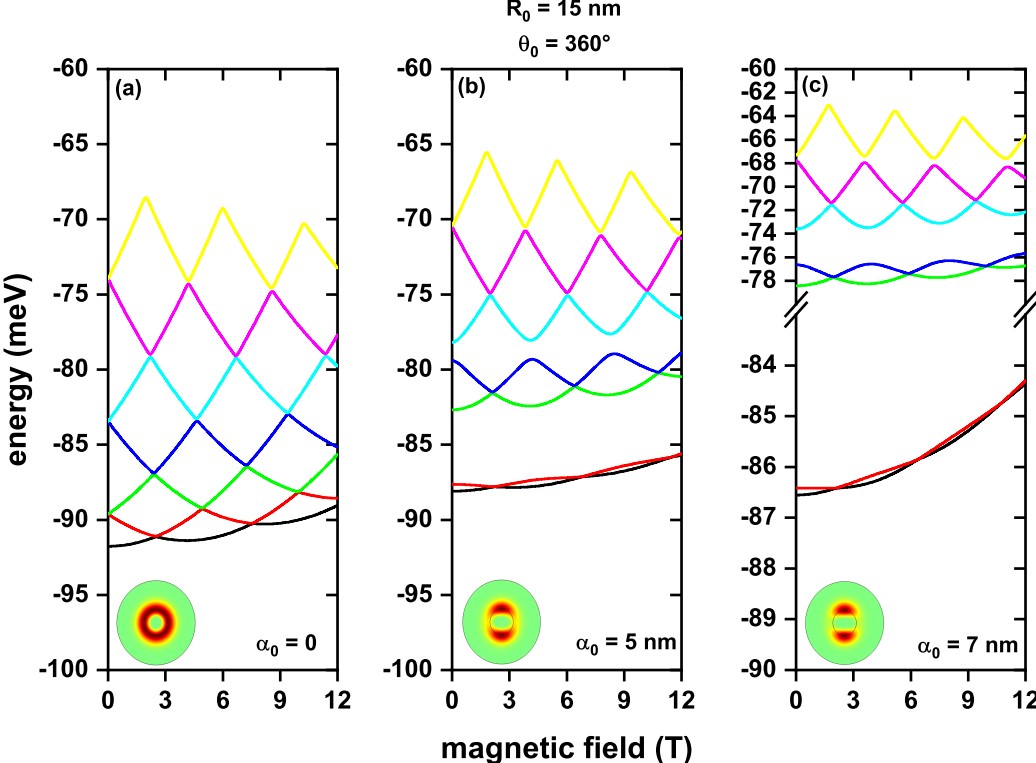

**Figure 3.** First seven electron energy levels confined in a GaAs quantum ring under the inversely quadratic potential, as a function of the magnetic field, without structural defects, and values of the non-resonant intense laser of $\alpha_0 = 0$ (**a**), $\alpha_0 = 5\,\text{nm}$ (**b**), and $\alpha_0 = 7\,\text{nm}$ (**c**). The box in each figure shows the ground state wavefunction to indicate the laser's effect on the structure.

Figure 4 displays the first seven energy levels of an electron confined in a QR under the influence of an applied magnetic field and different values of the laser parameter, taking into account a structural defect of $10°$ in the structure's geometry. Figure 4a corresponds to the case without a laser parameter, where one may observe that the lower energy levels increased as the magnetic field strength augmented, indicating an increment in the confinement of the charge carrier. Additionally, no degenerate states were present, even when the applied magnetic field was zero, implying that including the structural defect introduced an asymmetry in the system, as expected. Note that the accidental degeneracies causing the crossings in energy levels were eliminated when considering this structural defect, resulting in energy curves without oscillating patterns and improved transition energies compared to the case without such a defect. Moreover, the oscillations of the energy levels also disappeared when the laser parameter was introduced to the system, as shown in Figure 4b,c. However, the energies were higher than in the case without the laser parameter (see Figure 4a) due to the states being less confined with the inclusion of the laser. We observed that the lower energy states tended to form pairs, and their separation reduced as the laser parameter increased, causing a shift towards lower energies between the ground state and the first excited state and between the ground state and the second excited state. The energies in Figure 4 are higher than in Figure 3 due to the increase in confinement produced by the structural defect. The boxes in Figure 4 display the ground state wavefunction for each case ($\alpha_0 = 0$, $\alpha_0 = 5\,\text{nm}$, and $\alpha_0 = 7\,\text{nm}$). It is worth noting that the wavefunction's effect can be noticed by extracting an angular sector in the geometry, leading to a non-uniform distribution of wave functions.

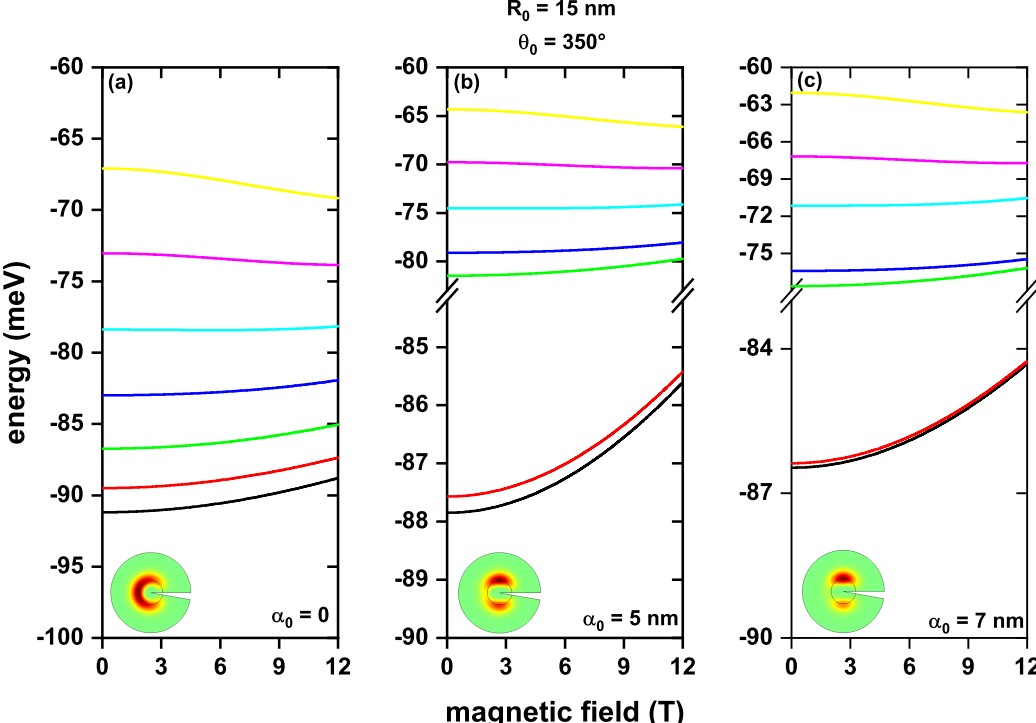

**Figure 4.** First seven electron energy levels confined in a GaAs quantum ring under the inversely quadratic potential, as a function of the magnetic field, with a structural defect of 10° in the geometry of the structure, and values of the non−resonant intense laser of $\alpha_0 = 0$ (**a**), $\alpha_0 = 5\,\text{nm}$ (**b**), and $\alpha_0 = 7\,\text{nm}$ (**c**). The box of each figure shows the ground state wavefunction to indicate the laser's effect on the structure.

Figure 5 presents the first seven energy levels of an electron confined in a QR as functions of the laser parameter for different values of the applied magnetic field. For all three cases (Figure 5a–c), an increase in the energy levels was observed as the value of the laser parameter augmented. This was because applying the intense non-resonant laser displaced the potential towards higher energies. Consequently, the states became less confined, and the energy increased. In Figure 5a, it is noted that the excited states were doubly degenerate for $\alpha_0 = 0$. However, the degeneracy was lost when the laser parameter was activated due to the symmetry breakdown induced in the system. It was also noticed that the separation between levels, induced by the laser parameter, decreased for the upper states. When the applied magnetic field was not equal to zero (Figure 5b,c), there was no degeneracy. On the other hand, if B was intense enough (Figure 5c), crossings in some states can be observed due to accidental degeneracy for high values of $\alpha_0$. It is essential to highlight the symmetry breaking a structural defect induces in the system. Consequently, a loss of degeneracy in the states was observed, even when there was no magnetic or laser field (see Figure 6). Additionally, compared to the case without a structural defect (Figure 5), an increase in energy was noted due to the removal of an angular sector of 10° in the geometry of the structure, which led to an increase in the confinement of the electron.

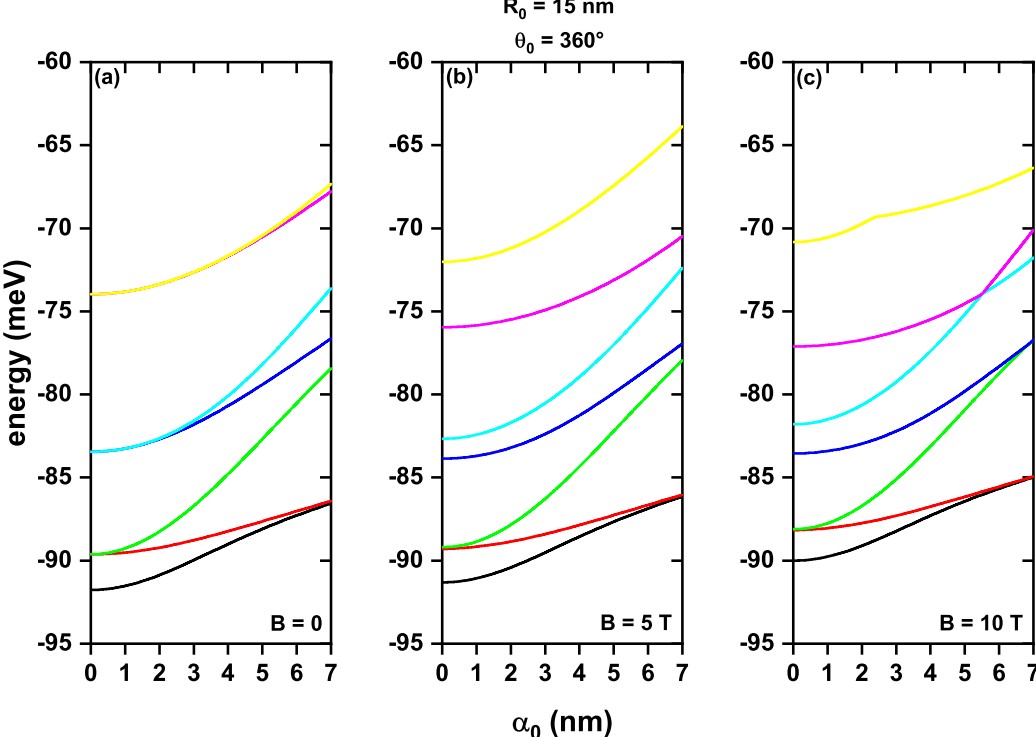

**Figure 5.** First seven electron energy levels confined in a GaAs quantum ring under the inversely quadratic potential, as a function of laser parameter $\alpha_0$, without a structural defect, and values of magnetic field of $B = 0$ (**a**), $B = 5\,\mathrm{T}$ (**b**), and $B = 10\,\mathrm{T}$ (**c**).

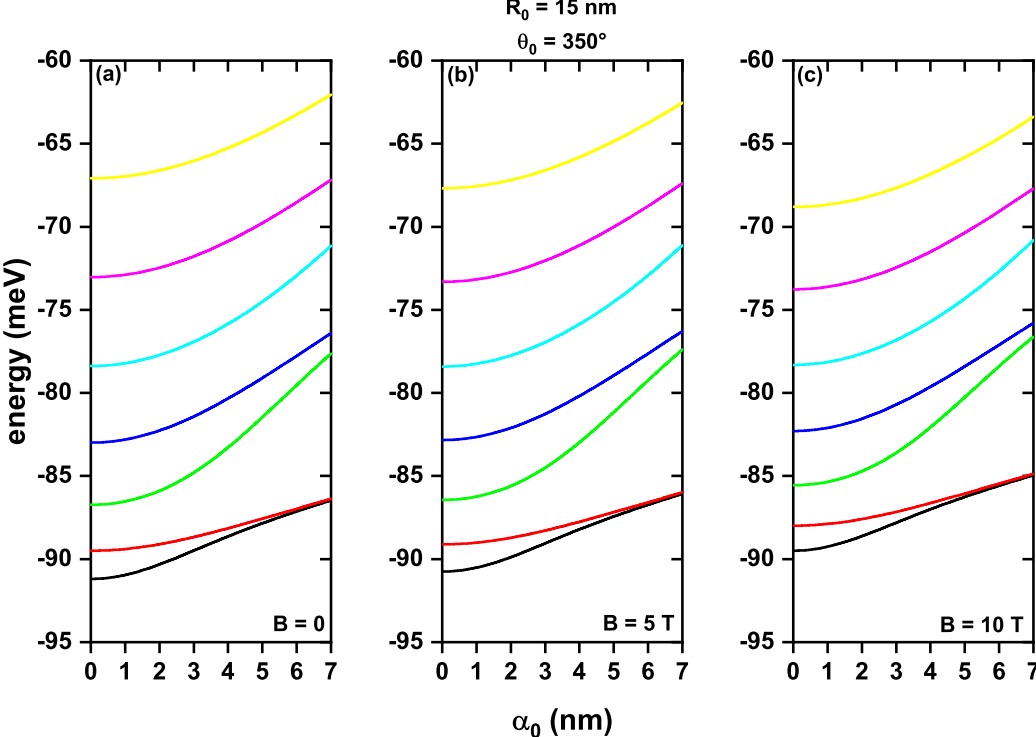

**Figure 6.** First seven electron energy levels confined in a GaAs quantum ring under the inversely quadratic potential, as a function of laser parameter $\alpha_0$, with structural defects, and values of magnetic field of $B = 0$ (**a**), $B = 5\,\mathrm{T}$ (**b**), and $B = 10\,\mathrm{T}$ (**c**).

Now, we focused our attention on studying the allowed transitions between the states examined in this work. To this end, Figure 7 presents the squares of the dipole moment matrix elements as functions of the applied magnetic field for two different laser parameter values, with right circular polarization—indicated by black circles—and left circular polarization—by empty red circles—when no structural defect is present in the structure. Figure 7a,b display the squares of the dipole moments for the transitions $1 \rightarrow 2$ with values of $\alpha_0$ of 5 nm and 7 nm, respectively. It was observed that $|M_{1j}|^2$ was non-zero for any value of the magnetic field due to the parity change in the wavefunctions between the ground state and the first excited state. Specifically, for $B = 0$, the wave function of the ground state had even parity, while that of the first excited state had odd parity. The dipole moment also exhibited an alternating pattern between left and right circular polarization. This led to one type of polarization predominating over the other at certain magnetic field values and an oscillating pattern between these two polarizations as a function of $B$. Generally, the magnitude of the dipole moment increased as the laser parameter rose, but the amplitude of the oscillation of $|M_{1j}|^2$ decreased for large values of $\alpha_0$. When examining the transition $1 \rightarrow 3$ (Figure 7c,d), the alternating pattern of the dipole moment as a function of the magnetic field was maintained. Specifically, for $B = 0$, $|M_{1j}|^2$ began to increase for right circular polarization and decrease for left circular polarization. However, the oscillating pattern of the dipole moment did not have a well-defined shape in this case, and the range of values of $|M_{1j}|^2$ was lower when $\alpha_0 = 7$ nm compared to when $\alpha_0 = 5$ nm. The vertical window of the values in Figure 7c occupied a range of 60 nm$^2$, while in Figure 7d, the range was 25 nm$^2$.

Figure 8 presents the squares of the dipole moment matrix elements as functions of the applied magnetic field, taking into account a structural defect of $10°$ in the geometry of the structure (as shown in Figure 2b), for two values of the laser parameter and with right circular polarization (black circles) and left circular polarization (empty red circles). It was evident that, when a structural deformation in the structure was considered, the dipole moment lost its oscillatory character with $B$ for all cases, showing a decreasing behavior as the magnetic field increased. This reduction can be associated with a decrease in the overlap of the wave functions of the involved states. Additionally, there was a general reduction in the magnitude of the dipole moment for both polarizations as the laser parameter increased. This behavior was due to a more intense laser field distributing the wavefunctions in a smaller region of space, leading to a decrease in their overlap with the $x$ or $y$ linear terms. It is worth noting that $M_{1j}^2$ was larger than the $1 \rightarrow 3$ element, as noticed from the pairs of Figure 8a–d. This was because the ground state and the first excited state retained well-defined even and odd parities, respectively. In contrast, the second excited state did not have a well-defined parity under the laser field influence.

Tables 1 and 2 show the results of the squares of the dipole moments $|M_{12}|^2$ and $|M_{13}|^2$ and their respective transition energies $E_{12}$ and $E_{13}$ for four different magnetic field values and intense laser parameters of 5 nm and 7 nm, with right and left circular polarization; without and with a structural defect, respectively. Equation (4) was used to obtain the linear absorption coefficient as a function of the energy of the incident photon; note that there was no magnetic field value where the square of the dipole moment was zero, so two peaks would appear at the absorption curves for each value of $B$.

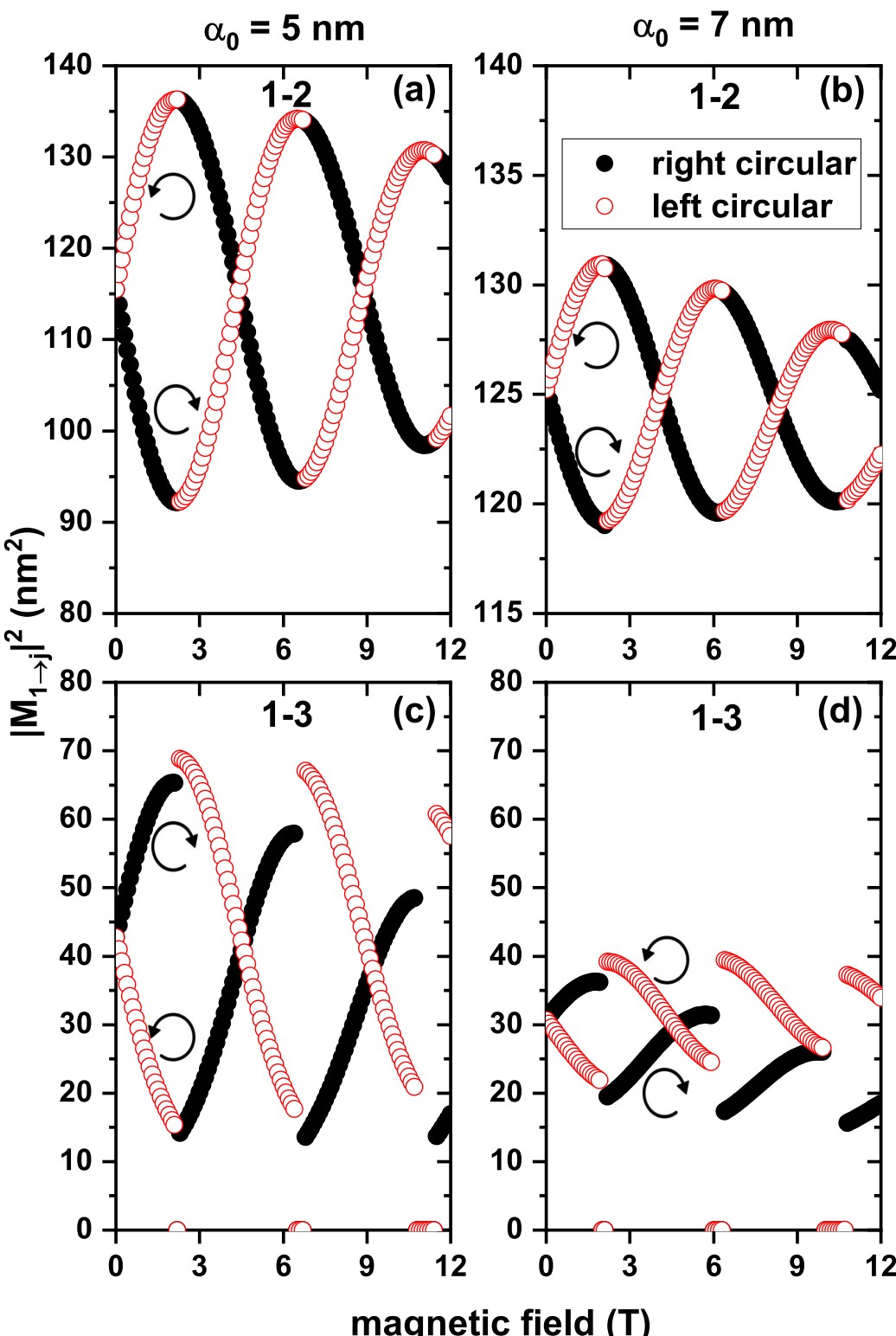

**Figure 7.** Squared dipole moments as functions of the applied magnetic field for an electron confined in a GaAs quantum ring under the inversely quadratic potential, without structural defects and with $\alpha_0 = 5\,\text{nm}$ (**a,c**) and $\alpha_0 = 7\,\text{nm}$ (**b,d**).

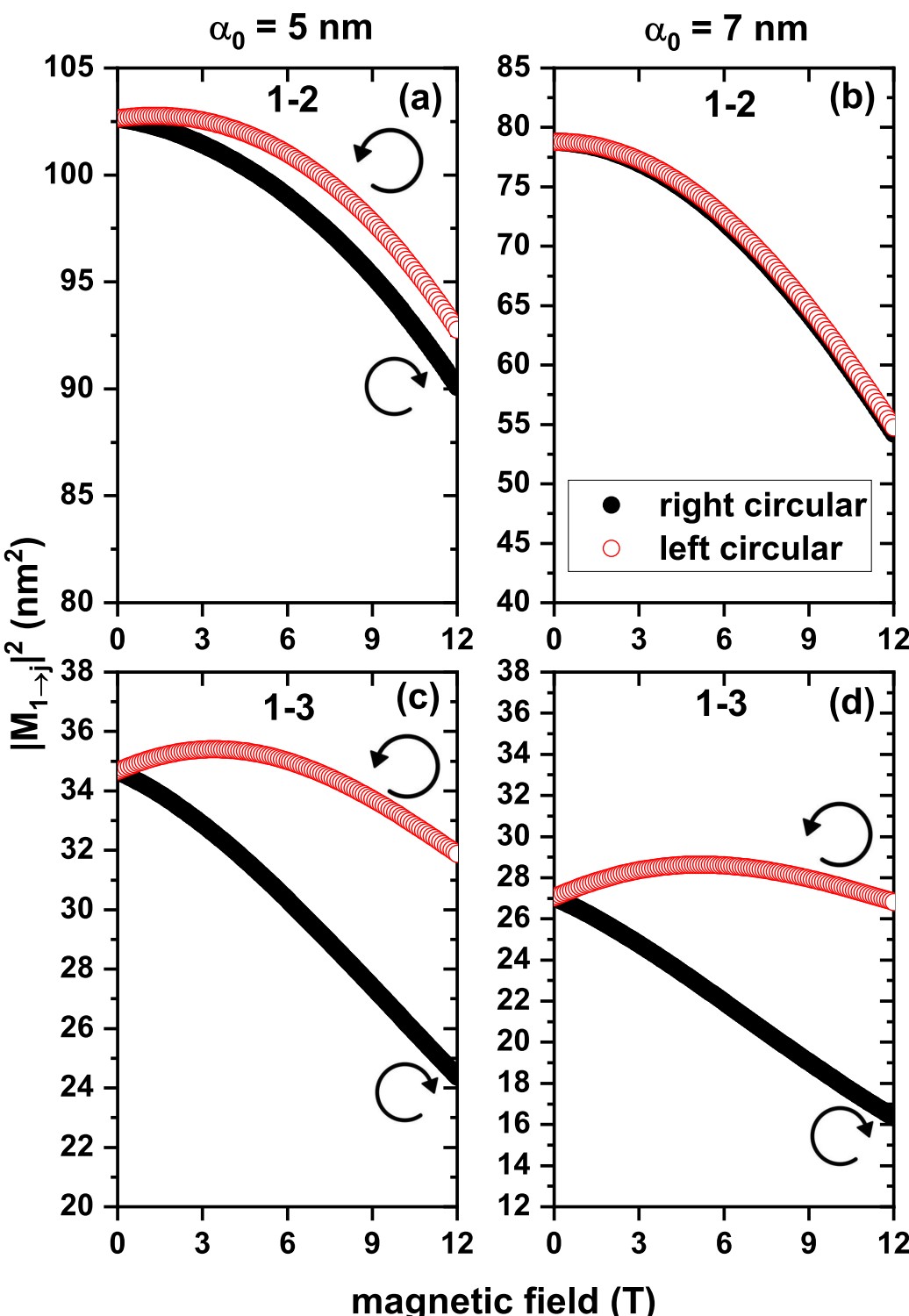

**Figure 8.** Squared dipole moments as functions of the applied magnetic field for an electron confined in a GaAs quantum ring under the inversely quadratic potential, with structural defects and with $\alpha_0 = 5\,\text{nm}$ (**a**,**c**) and $\alpha_0 = 7\,\text{nm}$ (**b**,**d**).

**Table 1.** Values of the dipole moments of the $|M_{12}|^2$ and $|M_{13}|^2$ transitions and their corresponding transition energies $E_{12}$ and $E_{13}$, for different values of the laser non-resonant parameter, without structural defects.

| $\theta_0 = 360°$ | Right Circular Polarization | | | $\alpha_0$ (nm) | Left Circular Polarization | | |
|---|---|---|---|---|---|---|---|
| | B (T) | | | | B (T) | | |
| | 0 | 2.1 | 8.7 | | 0 | 2.1 | 8.7 |
| $|M_{12}|^2$ (nm$^2$) | 115.51 | 92.28 | 117.71 | | 115.51 | 136.29 | 112.98 |
| $|M_{13}|^2$ (nm$^2$) | 42.79 | 65.36 | 32.37 | | 42.79 | 15.36 | 45.96 |
| $E_{12}$ (meV) | 0.46 | 0.05 | 0.34 | 5 | 0.46 | 0.05 | 0.34 |
| $E_{13}$ (meV) | 5.42 | 6.28 | 5.22 | | 5.42 | 6.28 | 5.22 |
| $|M_{12}|^2$ (nm$^2$) | 125.22 | 130.88 | 123.10 | | 125.22 | 119.23 | 125.67 |
| $|M_{13}|^2$ (nm$^2$) | 30.70 | 19.50 | 24.71 | | 30.70 | 39.22 | 30.81 |
| $E_{12}$ (meV) | 0.14 | 0.01 | 0.10 | 7 | 0.14 | 0.01 | 0.10 |
| $E_{13}$ (meV) | 8.12 | 8.57 | 7.92 | | 8.12 | 8.57 | 7.92 |

**Table 2.** Values of the dipole moments of the $|M_{12}|^2$ and $|M_{13}|^2$ transitions and their corresponding transition energies $E_{12}$ and $E_{13}$, for different values of the laser non-resonant parameter, with a structural defect.

| $\theta_0 = 350°$ | Right Circular Polarization | | | $\alpha_0$ (nm) | Left Circular Polarization | | |
|---|---|---|---|---|---|---|---|
| | B (T) | | | | B (T) | | |
| | 0 | 8 | 12 | | 0 | 8 | 12 |
| $|M_{12}|^2$ (nm$^2$) | 102.64 | 96.66 | 90.09 | | 102.64 | 99.06 | 92.77 |
| $|M_{13}|^2$ (nm$^2$) | 34.66 | 28.40 | 24.39 | | 34.66 | 34.23 | 31.88 |
| $E_{12}$ (meV) | 0.28 | 0.23 | 0.18 | 5 | 0.28 | 0.23 | 0.18 |
| $E_{13}$ (meV) | 6.39 | 6.12 | 5.88 | | 6.39 | 6.12 | 5.88 |
| $|M_{12}|^2$ (nm$^2$) | 78.78 | 67.12 | 54.23 | | 78.78 | 67.67 | 54.75 |
| $|M_{13}|^2$ (nm$^2$) | 27.05 | 19.99 | 16.35 | | 27.05 | 28.25 | 26.82 |
| $E_{12}$ (meV) | 0.09 | 0.07 | 0.06 | 7 | 0.09 | 0.07 | 0.06 |
| $E_{13}$ (meV) | 8.83 | 8.49 | 8.14 | | 8.83 | 8.49 | 8.14 |

We plot, in Figure 9, the linear optical absorption coefficient as a function of the incident photon energy for magnetic fields of 0, 2.1, and 8.7 T and laser parameters of 5 and 7 nm, with right and left circular polarization and without structural defects in the system. The choice of these specific values of *B* enabled us to observe differences in the positions of the resonant peaks since, in some cases, the peaks were obtained at very similar positions. As seen, the resonant peak positions did not exhibit a monotonic behavior as *B* increased due to the oscillating nature of the energy levels as a function of the magnetic field. Note that a similar pattern was also observed concerning the magnitude of the resonant peaks as the magnetic field increased since the intensity of the peaks was primarily related to $\omega |M_{1j}|^2$. Consequently, the product of the transition energy with the square of the dipole moment followed this oscillation pattern as the magnetic field increased. It is worth noting that the absorption peaks at the highest energies corresponded to the $1 \rightarrow 3$ transition since the energy spacing between the ground and the first excited state was larger than the $1 \rightarrow 2$ transition, as demonstrated in Figure 3b,c. Furthermore, the transition energies for Figure 9a,c were the same, as well as those of Figure 9b,d.

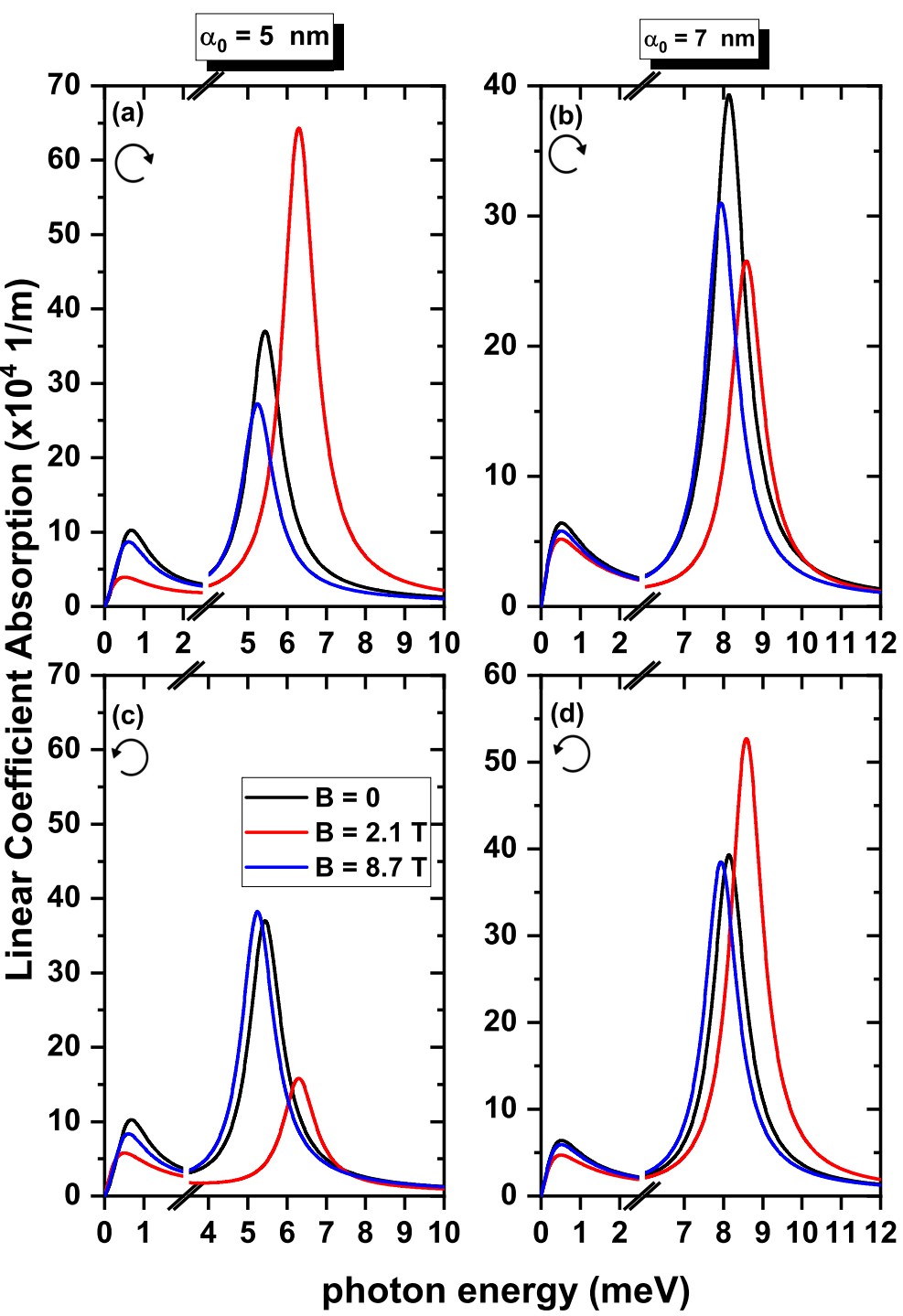

**Figure 9.** The linear optical absorption coefficient is a function of the photon energy for the electron confined in a GaAs quantum ring under the inversely quadratic potential, without considering the structural defect in the structure's geometry. The calculations were performed with right circular polarization (**a**,**b**) and left circular polarization (**c**,**d**).

Finally, Figure 10 depicts the linear absorption coefficient as a function of the incident photon energy for different applied magnetic fields (0, 8, and 12 T) and laser parameters (5 and 7 nm), with right and left circular polarization in the GaAs QR with a structural defect. The selected values of magnetic fields were chosen to extend the range of energies at which the resonant peaks were observed. Compared to Figure 9, the resonant peaks in Figure 10 show a monotonic behavior and shift towards lower energies as *B* increased for both polarizations and laser fields. This behavior was due to the elimination of the

oscillating character of the energy curves and the reduction of the transition energies $1 \to 2$ and $1 \to 3$ caused by including a structural defect, as shown in Figure 2. In addition, the intensity of the peaks for $B = 0$ in Figure 10 for transitions $1 \to 2$ and $1 \to 3$ was the same since the value of $|M_{ij}|^2$ for both polarizations was equal at this magnetic field. However, a general decrease in the absorption intensity was observed as $B$ increased for Figure 10a–c, due to the dipole moment decreasing as a function of $B$, as shown in Figure 8a–c. An exception was made for Figure 10d, where there was not much difference in the value of $|M_{ij}|^2$ between magnetic fields of zero and 9 T.

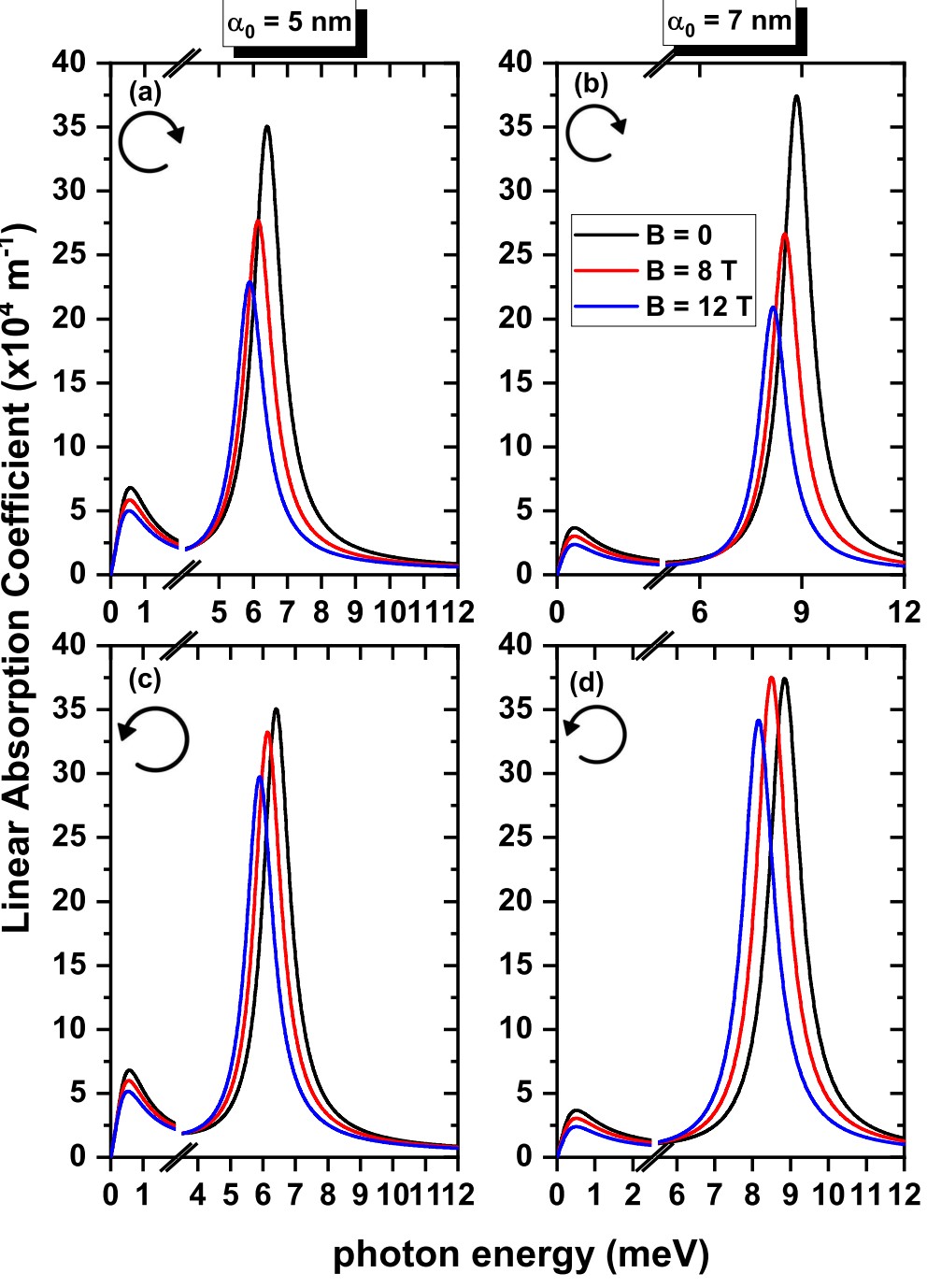

**Figure 10.** The linear optical absorption coefficient as a function of the photon energy for the electron confined in a GaAs quantum ring under the inversely quadratic potential, considering a structural defect of 10°. The calculations were performed right circular polarization (**a,b**) and left circular polarization (**c,d**).



## 4. Conclusions

We conducted a study to investigate the impact of a magnetic field, an intense non-resonant laser field, and a structural defect on the electronic and optical properties of an electron confined in a GaAs quantum ring with an inversely quadratic confinement potential. To obtain the energies and eigenfunctions of the system, we solved the Schrödinger equation in two dimensions using the finite-element method. Our findings are summarized as follows: (i) the axial symmetry of the excited states was broken by both the laser field and the structural defect, and for larger values of the laser intensity parameter, the states were almost degenerate; (ii) the wavefunction of the ground state showed two well-defined and isolated lobes when there was an increase in the parameter that controlled the laser intensity in the system; (iii) the inclusion of a structural defect eliminated the oscillatory character in the energy curves and increased the transition energy between states $1 \rightarrow 2$ and $1 \rightarrow 3$, compared to when the defect was not considered; finally, (iv) the transition $1 \rightarrow 3$ was the most-significant contribution to both the intensity and position of the resonant peak when the two external effects were considered, resulting in a significant improvement in the energy of the resonant peak compared to the case where neither of the effects were present.

**Author Contributions:** J.C.L.-G.: conceptualization, methodology, software, formal analysis, investigation, writing; R.G.T.-N.: methodology, software; A.L.M. and J.A.V.: formal analysis, investigation, supervision, writing; M.E.M.-R. and C.A.D.: formal analysis, writing. All authors have read and agreed to the published version of the manuscript.

**Funding:** The authors are grateful to the Colombian Agencies: CODI-Universidad de Antioquia (Estrategia de Sostenibilidad de la Universidad de Antioquia and projects "Propiedades magneto-ópticas y óptica no lineal en superredes de Grafeno", "Estudio de propiedades ópticas en sistemas semiconductores de dimensiones nanoscópicas", "Propiedades de transporte, espintrónicas y térmicas en el sistema molecular ZincPorfirina", and "Complejos excitónicos y propiedades de transporte en sistemas nanométricos de semiconductores con simetría axial") and Facultad de Ciencias Exactas y Naturales-Universidad de Antioquia (ALM and CAD exclusive dedication projects 2022–2023). MEMR thanks CONACYT of Mexico for its support through Subsidy CB/2017-2018 A1-S-8218.

**Institutional Review Board Statement:** Not applicable.

**Informed Consent Statement:** Not applicable.

**Data Availability Statement:** No new data were created nor analyzed in this study. Data sharing is not applicable to this article.

**Conflicts of Interest:** The authors declare no conflict of interest.

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
