# Peer review of "Influence of a Non-Resonant Intense Laser and Structural Defect on the Electronic and Optical Properties of a GaAs Quantum Ring under Inversely Quadratic Potential"

_condensedmatter, doi:10.3390/condmat8020052_

Round 1

Reviewer 1 Report

The paper describes the theoretical study of the opto-electronic properties of a GaAs nano-ring. More precisely, it describes the intersubband transitions (from the fundamental to the first and second excited states) in the ring. This is the first problem in this paper: authors never explicitely say that they will calculate intersubband transitions. It could have been interband transitions, where the band structure of GaAs would play an important role. In intersubband transition, the band stsructure of GaAs plays a minor role (the effective mass is the most important parameter).

The introduction does not explain the context of this calculation. It remains very confused. It s hard to understand whether electronic states will be discussed, or whether it will be about optical and guided modes properties. The introduction is a list of references where rings have been studied, without explaining in details what it is about and more importantly without giving the link with the present paper. Hence, the introduction does not allow to justify the present calculation and present paper, which looks like an academic exercise. The introduction also contains some repetitions.

Some sentences remain unclear or even confused. For instance authors wrote : Such fields excite the charge carrier in the ring non-resonantly, meaning that the frequency of the laser field does not match the vibration frequency of the electron in the ring. What do author mean by vibration frequency of the electrons in the ring ? intersubband transitions from the fundamental to excited states ? or band to band transitions ? is this just free electron absorption ? authors must be more explicit. Another example is a mention of "two dimensional quantum dot". What does it mean ? a QD is 0D object with confinement in all directions.

Overall the lack of accuracy in the physics related to the structures described in the introduction or later in the text, is detrimental to the relevance of the paper.

Nevertheless, the calculation in itself and the results seem to be interesting and may deserve publication, once the deficiencies mentioned above will be corrected.

minor errors

Author Response

Referee 1

The Referee:

The paper describes the theoretical study of the opto-electronic properties of a GaAs nano-ring. More precisely, it describes the intersubband transitions (from the fundamental to the first and second excited states) in the ring. This is the first problem in this paper: authors never explicitely say that they will calculate intersubband transitions. It could have been interband transitions, where the band structure of GaAs would play an important role. In intersubband transition, the band stsructure of GaAs plays a minor role (the effective mass is the most important parameter).

Our reply:

The authors would like to express our most sincere gratitude to the Referee for his/her evaluation of our article. His/her opinions and comments have helped us to substantially improve our work.

In the third paragraph of the Introduction section we have added the following comment, which we believe clarifies the type of transitions considered in our study:

The present work is devoted, in first place, to investigate the spectrum of energy levels in a semiconductor two-dimensional QR whose design includes a particular description of the confining potential function as well as the possibility of having a structural defect. In addition, the study includes an analysis of the influence of external electromagnetic probes such as a static magnetic field and a non-resonant intense laser field on electron states. It implies to determine the solution of the conduction-band effective mass equation under the stated conditions. Secondly, the information from the allowed quantum states will serve as the basis for the calculation of one of the possible optical responses associated to transitions between them: the coefficient of intersubband light absorption (here, as it is customary, we shall keep the term "intersubband", although no band is actually present in the electronic structure of the system).

Before Eq. (4), we clarify with the following text that we are dealing with intersubbanda transicions. The text reads:

When the system energies and wavefunctions have been obtained by solving Eq. (1), the features of intersubband transitions reflect in the linear optical absorption coefficient, which is evaluated from the following expression

The Referee:

The introduction does not explain the context of this calculation. It remains very confused. It s hard to understand whether electronic states will be discussed, or whether it will be about optical and guided modes properties. The introduction is a list of references where rings have been studied, without explaining in details what it is about and more importantly without giving the link with the present paper. Hence, the introduction does not allow to justify the present calculation and present paper, which looks like an academic exercise. The introduction also contains some repetitions.

Our reply:

We thank the Referee for his/her comment. We completely agree with the point of view of the Referee. We have modified the Introduction section leaving in it the citation of references that are very important to establish where our research fits. We have clarified in detail the novelty of our work. We have established the connection between our research and what is known in the literature.

The Referee:

Some sentences remain unclear or even confused. For instance authors wrote: Such fields excite the charge carrier in the ring non-resonantly, meaning that the frequency of the laser field does not match the vibration frequency of the electron in the ring. What do author mean by vibration frequency of the electrons in the ring? intersubband transitions from the fundamental to excited states? or band to band transitions? is this just free electron absorption? authors must be more explicit. Another example is a mention of "two dimensional quantum dot". What does it mean? a QD is 0D object with confinement in all directions.

Our reply:

We thank the Referee for his/her comment. The sentence which reads:

Non-resonant intense laser fields offer another way to control the properties of QRs. Such fields excite the charge carrier in the ring non-resonantly, meaning that the frequency of the laser field does not match the vibration frequency of the electron in the ring. This produces changes in the confinement potential, and due to the field's intensity, a non-linear interaction occurs between the confined electron and the field.

has been replaced by the following sentence:

Non-resonant intense laser fields offer another way to control the properties of QRs. Such fields excite the charge carrier in the ring non-resonantly, meaning that the laser field's energy does not match any of the possible energy transitions of the confined electron in the ring. The average temporal effect of the non-resonant laser field is to produce changes in the confinement potential, which confines the electron into the heterostructure region.

In the third paragraph of the Introduction section we have added the following comment, which we believe clarifies the type of transitions considered in our study:

The present work is devoted, in first place, to investigate the spectrum of energy levels in a semiconductor two-dimensional QR whose design includes a particular description of the confining potential function as well as the possibility of having a structural defect. In addition, the study includes an analysis of the influence of external electromagnetic probes such as a static magnetic field and a non-resonant intense laser field on electron states. It implies to determine the solution of the conduction-band effective mass equation under the stated conditions. Secondly, the information from the allowed quantum states will serve as the basis for the calculation of one of the possible optical responses associated to transitions between them: the coefficient of intersubband light absorption (here, as it is customary, we shall keep the term "intersubband", although no band is actually present in the electronic structure of the system).

We thank the Referee for his/her question. We want to highlight that the absorption coefficient we calculate in this study is associated with transitions between states of an electron confined in the quantum ring. Given the confinement in all directions of space and that the dimensions of the heterostructure are of the order of magnitude of the effective Bohr radius, quantum effects become visible. From the solution of the Schrödinger equation, a set of energy states appears discreet. It is precisely between these discrete states that the absorption of photons is considered. Changes in the size and shape of the heterostructure induce changes in the energy spectrum. So, we can conclude that the absorption process is for confined electrons. This is not a free electron process.

The title of Ref. [27] is: “Impact of a topological defect and Rashba spin-orbit interaction on the thermo-magnetic and optical properties of a 2D semiconductor quantum dot with Gaussian confinement”. Actually, in Fig. 1 of Ref. [27], what appears there is a disc with a very small height compared to its radius. The authors concentrate their entire study on a two-dimensional problem, ignoring the effect of height, which is seen in Eq. (11) of Ref. [11]. We agree with the Referee, the quantum dots are zero-dimensional systems, which indicates that there is confinement in the three directions of space. We understand in the title of the article that the term 2D refers to the fact that the authors use a differential equation of eigenvalues in polar coordinates (r, q). In the quote we make of the article, we have wanted to respect the title the authors have given to their work.

The Referee:

Overall the lack of accuracy in the physics related to the structures described in the introduction or later in the text, is detrimental to the relevance of the paper.

Nevertheless, the calculation in itself and the results seem to be interesting and may deserve publication, once the deficiencies mentioned above will be corrected.

Our reply:

We hope that the Referee finds satisfactory our answers to his/her questions and the changes we have made to the manuscript. We believe that the Referee has guided us clearly and forcefully to improve the quality of our work. We hope that in the opinion of the Referee the article can be accepted for publication in the Condensed Matter journal.

Reviewer 2 Report

Title: Influence of a non-resonant intense laser and topological defect on the electronic and optical properties of a GaAs quantum ring under inversely quadratic potential
Overview and general recommendation:
The manuscript reports the impact of a non-resonant intense laser, topological defect, and magnetic field on the electronic and optical properties of a simple GaAs quantum ring under the inverse quadratic Hellmann potential, using the effective mass and parabolic band approximations. By solving the Schrödinger equation in two dimensions using the finite element method. and a series of disciplines is conducted. However, before I can recommend the article for publication, I have the following comments:
Comments:

1. 49-51 lines “Overall, the findings of these studies demonstrate the significant impact that external factors can have on the properties of quantum rings, which is essential for optimizing these structures for various applications”. Maybe some reference or examples for applications will help with this clarification job.

2. 111-113 lines: “As mentioned, the confinement potential is modeled using the inversely quadratic Hellmann potential, which is given by [30].”, maybe explain detail will be better.

3. 137-138 lines: “Thus, the angular amplitude of the topological defect introduced in the problem is of 10°.”

a) What is the reason for the value of ten degree? or the numbers have been picked by experience, or any specific reasons?

b) How about the smaller and larger degree?

c) How to define and understand topology defects? For what physical quantity topology?

4. 247-248 lines: what is the root cause behind “the range of values of |M1j|2 is lower when a0 = 7 nm compared to when a0 = 5 nm”.?

5. The left and right rotation markings in Figure 7 are not obvious, Suggest consistency with Figure 8. What is the reason for there is a different point near 11 in Figure 7b?

6. Any definitions and mathematical expressions corresponding transition energies E12 and E13 could help readers to understand them better, how are the specific values calculated? Transition energies E12 and E13 in Table 1 and Table 2 have no difference for same a0, or maybe the question is: if one needs to design a QR, how does him/her pick the laser parameters? 

7. In figure 9, How to understand the phenomenon of the linear optical absorption coefficient values under different B, what is the reason behind the pattern of peak value in the figure is not obvious?

8. Finally, it is suggested that the article discuss what guiding significance these conclusions will have for the future, or what specific problems will be solved.

Author Response

Referee 2:

The Referee:

The manuscript reports the impact of a non-resonant intense laser, topological defect, and magnetic field on the electronic and optical properties of a simple GaAs quantum ring under the inverse quadratic Hellmann potential, using the effective mass and parabolic band approximations. By solving the Schrödinger equation in two dimensions using the finite element method. And a series of disciplines is conducted. However, before I can recommend the article for publication, I have the following comments:

Our reply:

The authors would like to express our most sincere gratitude to the Referee for his/her evaluation of our article. His/her opinions and comments have helped us to substantially improve our work.

The Referee:

  1. 49-51 lines “Overall, the findings of these studies demonstrate the significant impact that external factors can have on the properties of quantum rings, which is essential for optimizing these structures for various applications”. Maybe some reference or examples for applications will help with this clarification job.

Our reply:

We have added the following reference:

“Fomin, V. M. \textit{Physics of Quantum Rings}, 2nd ed.; Springer, Dresden, Germany, 2014.”

This is an excellent book on quantum rings where it is detailed from the theory, the experimental part, the different growth techniques, and even the multiple technological applications of these systems.

The Referee:

  1. 111-113 lines: “As mentioned, the confinement potential is modeled using the inversely quadratic Hellmann potential, which is given by [30].”, maybe explain detail will be better.

Our reply:

We thank the Referee for his/her comment. In the text that follows Eq. (2), we have presented full details on the confinement potential that we have used in this study. In Fig. 1, also we present the confining potential without and with the effects of non-resonant intense laser radiation with linear polarization along the x-axis.

In the paragraph that follows Eq. (2), we have added the following text:

From a practical point of view, this type of potential is achieved by an intentional variation of the aluminum concentration along the radial direction of the heterostructure. This is a challenging issue and quite complex to achieve. However, it is important to highlight that these non-abrupt variations of the confinement potentials occur in the interdiffusion process between regions of wells and barriers when these are intended to be built abruptly. At this point, the model that we present here takes on its greatest relevance.

The Referee:

  1. 137-138 lines: “Thus, the angular amplitude of the topological defect introduced in the problem is of 10°.”
  2. a) What is the reason for the value of ten degree? or the numbers have been picked by experience, or any specific reasons?
  3. b) How about the smaller and larger degree?
  4. c) How to define and understand topology defects? For what physical quantity topology?

Our reply:

We thank the Referee for his/her comments and questions.

First, we want to comment that we have changed the words “topological defect” to “structural defect” throughout the manuscript. In our case, we are considering a structural defect in the form of a circular sector cut. The value of the angular defect of 10° has been chosen as an illustrative example for developing the manuscript.

The physics that is presented for greater or lesser values of said angle remains mostly the same concerning what we are already showing in this manuscript. When considering a larger angle, the area of the confinement region is reduced, and for that reason, the energies of the confined states increase, and the asymmetry of the wave functions also increases. When considering smaller angles, the opposite situation occurs; the area of the confinement region increases, and the energies of the confined states decrease. Even in the case in which the angle of the defect tends to zero, in any case, a potential barrier will appear in the system along the two radial lines of the defect, and therefore, the breaking of the azimuthal symmetry of the system will always be present. These defects, which are responsible for the symmetry breaking of the system, will give rise to optically allowed transitions between confined states for certain linear polarizations of the incident resonant radiation. A complete study with variations in the defect's angle and other different types of defects is in progress and will be published elsewhere.

The defects we are reporting here allow us to understand the physics of nanoflakes, sheet-like structures with various geometries and sizes that arise in the growth process of quasi-two-dimensional structures.

The Referee:

  1. 247-248 lines: what is the root cause behind “the range of values of |M1j|2is lower when a0= 7 nm compared to when a0 = 5 nm”.?

Our reply:

In order to clarify, in the final part of the paragraph where Fig. 7 is discussed we have added the following text:

The vertical window of the values in Fig. 7(c) occupies a range of 60\,nm$^2$ while in Fig. 7(d) the range is 25\,nm$^2$.

The Referee:

  1. The left and right rotation markings in Figure 7 are not obvious, Suggest consistency with Figure 8. What is the reason for there is a different point near 11 in Figure 7b?

Our reply:

We have corrected Fig. 7 to improve presentation and provide clarity on left and right circular polarization. In Fig, 8 we have added the labels (a), (b), (c), (d). In Fig. 7(b) we have detected an error in the data when exporting it to the table. We have corrected the error in the table.

The Referee:

  1. Any definitions and mathematical expressions corresponding transition energies E12 and E13 could help readers to understand them better, how are the specific values calculated? Transition energies E12 and E13 in Table 1 and Table 2 have no difference for same a0, or maybe the question is: if one needs to design a QR, how does him/her pick the laser parameters?

Our reply:

After Eq. (4) in the revised version of the manuscritp we write: “$E_{12}=E_2-E_1$ and $E_{13}=E_3-E_1$”.

Tables 1 and 2 show that evidently quantum rings under the effects of intense non-resonant lasers are polarization detectors for incident radiation. This occurs fundamentally through the magnitude of the resonant peaks of the absorption coefficients, which change with the magnitude of $\alpha_0$ and the applied magnetic field through the magnitude of the dipole moment. Note that $\alpha_0$ is proportional to the square of the dipole moment.

The Referee:

  1. In figure 9, How to understand the phenomenon of the linear optical absorption coefficient values under different B, what is the reason behind the pattern of peak value in the figure is not obvious?

Our reply:

According to Eq. (4), in this study we are considering the absorption coefficient for transitions from the ground state to the first two excited states. Each transition contributes a Lorentzian function, whose resonant peak is located at the energy of the photon close to the considered transition energy. The peak is slightly offset since the effect of the damping parameter ($\Gamma$) is included in the problem. Since in the problem we are considering a sum of two transitions, see Eq. (4), then for each set of parameters the absorption coefficient presents two Lorentzian structures. A structure on the left side, corresponding to the lowest energy transition, E21, and one on the right side, corresponding to the highest energy transition, E31. The magnitude of each resonant peak in the Lorentzians is proportional to the product of the transition energy and the square of the dipole moment. This explains the reason why for each set of parameters the peak on the right (with higher transition energy) is always higher than the corresponding one on the left (with lower transition energy). The magnitude of the resonant peaks in Figs. 9 and 10 are essentially dominated by the transition energy.

The Referee:

  1. Finally, it is suggested that the article discuss what guiding significance these conclusions will have for the future, or what specific problems will be solved.

Our reply:

We have improved the introduction of the article to clarify what is new in this work and how it connects with the available literature.

We hope that the Referee finds satisfactory our answers to his/her questions and the changes we have made to the manuscript. We believe that the Referee has guided us clearly and forcefully to improve the quality of our work. We hope that in the opinion of the Referee the article can be accepted for publication in the Condensed Matter journal.

Round 2

Reviewer 1 Report

Previous comments from my side have been addressed in this revised version. Paper can be accepted.

Reviewer 2 Report

The authors have addressed all my concerns. I recommend it for publication.